# Enhanced Photocatalysis of Black TiO_2_/Graphene Composites Synthesized by a Facile Sol–Gel Method Combined with Hydrogenation Process

**DOI:** 10.3390/ma15093336

**Published:** 2022-05-06

**Authors:** Zhaoqing Li, Zhufeng Liu, Xiao Yang, Annan Chen, Peng Chen, Lei Yang, Chunze Yan, Yusheng Shi

**Affiliations:** State Key Laboratory of Materials Processing and Die & Mould Technology, School of Materials Science and Engineering, Huazhong University of Science and Technology, Wuhan 430074, China; lizhq84@hust.edu.cn (Z.L.); d201980300@hust.edu.cn (Z.L.); yangxiao@hust.edu.cn (X.Y.); nnanchennuaa@hust.edu.cn (A.C.); chenpengoppo@163.com (P.C.); jackyejackye@163.com (L.Y.); shiyusheng@hust.edu.cn (Y.S.)

**Keywords:** titanium dioxide, graphene, photocatalysis, hydrogenation, Ti^3+^ self-doping

## Abstract

In this study, in situ TiO_2_ was grown on the surface of graphene by a facile sol–gel method to form black TiO_2_/graphene composites with highly improved photocatalytic activity. The combination of graphene and TiO_2_ was beneficial to eliminate the recombination of photogenerated electron holes. The self-doping Ti^3+^ was introduced, accompanied by the crystallization of amorphous TiO_2_, during the hydrogenation process. Consequently, the narrowed bandgap caused by self-doping Ti^3+^ enhanced the visible light absorption and thus made the composites appear black. Both of them improved the photocatalytic performance of the synthesized black TiO_2_/graphene composites. The band structure of the composite was analyzed by valence band XPS, revealing the reason for the high visible light catalytic performance of the composite. The results proved that the black TiO_2_/graphene composites synthesized show attractive potential for applications in environmental and energy issues.

## 1. Introduction

Titanium dioxide (TiO_2_) with good photoelectric properties is generally regarded as an important photocatalyst [1,2]. Typically, TiO_2_ has four crystal phases, in which the anatase and rutile TiO_2_ are the most common and have been extensively investigated due to their excellent photoactivity [3,4]. However, their large band gap (rutile at ~3.0 eV and anatase at ~3.2 eV) severely limits the activity to the ultraviolet (UV) region of light. Typically, less than 5% of the entire solar energy is used for TiO_2_ photocatalysts [5,6]. Moreover, the photogenerated electron-hole pairs in TiO_2_ will recombine during the conduction process instead of participating in the photocatalytic reaction. This recombination of the electron hole would reduce the quantum efficiency that weakens the photocatalytic efficiency [4]. Therefore, great efforts have been made to shorten the absorption range of light and reduce the recombination of the photogenerated electron holes.

Over decades, the improvement of the photocatalytic activity of TiO_2_ has typically been achieved through appropriate structural design, synthesis of metal and non-metal element doping and semiconductor composite materials [7,8,9,10]. In recent years, some nanomaterials, such as carbon nanotubes, g-C3N4 and graphene with excellent electrical conductivity, have been recognized as attractive composite materials for improving the photocatalytic activity of TiO_2_ [11,12,13]. Notably, many reports have revealed that the excellent conductivity of graphene is conducive to the transfer of electrons. Therefore, the contact between TiO_2_ and graphene can significantly promote the recombination of photogenerated electron holes, thereby improving photocatalytic performance [14,15]. However, there are still some issues for the TiO_2_-graphene composites, such as their low visible light utilization [16,17]. In 2011, Chen et al. synthesized black TiO_2_ nanoparticles with a long wavelength absorption and substantial visible light photocatalytic activities by the hydrogenation method [18]. These remarkably changed properties made the black TiO_2_/graphene composite a promising candidate for the development of photocatalytic performance.

In this paper, in situ amorphous TiO_2_ was grown on the surface of graphene by a facile sol–gel method to form a series of black TiO_2_/graphene composites. The synthesized photocatalyst showed a narrow band gap and excellent photocatalytic performance. The influence of graphene on the photocatalytic activity of black TiO_2_/graphene composites has been systematically investigated by the degradation of methyl blue.

## 2. Materials and Methods

### 2.1. Synthesis of Black TiO_2_/Graphene Composites

Typically, the black TiO_2_/graphene composites were synthesized by a versatile sol–gel method, as shown in Figure 1. Tetrabutyl titanate (TBOT) (98.0%, Sinopharm Group Chemical Reagent Co., Ltd., Tianjin, China) and graphene (Strem Chemicals, Inc., Newburyport, MA, USA) were used as starting materials. Firstly, 5 mL TBOT, 250 mL C_2_H_5_OH (Eth), and various ratios of graphene were well mixed to obtain a mixture solution. The polyethylene glycol (PEG) was used as surfactant for enhancing the surface bonding between TiO_2_ and graphene. Then, the mixture solution was slowly dripped into a solution of 250 mL C_2_H_5_OH and 250 mL H_2_O while stirring. Through this process, the hydrolysis and polymerization of TBOT and H_2_O occurred to form a sol [19,20]. The overall reaction process is as follows,
(1)$${\mathrm{Ti}(\mathrm{OC}}_{4}H_{9}{)}_{4}+{2H}_{2}O\to {\mathrm{TiO}}_{2}+{4C}_{4}H_{9}\mathrm{OH}.$$

The above reaction was held for 1 h and then precipitated for 3 h. The precipitate was centrifuged and washed 3 times with ethanol. Subsequently, the amorphous TiO_2_/graphene composites were obtained after drying at 80 °C for 6 h and calcining at 200 °C for 2 h in air. As shown in Figure 1, the obtained composites were gradually darkened in color with various graphene contents (1 wt%, 5 wt%, 10 wt%, 15 wt%). Finally, the amorphous TiO_2_/graphene composites were calcined in H_2_ flow at 500 °C for 2 h under atmospheric pressure. Then the black TiO_2_/graphene composites (denoted as BTG-1, BTG-5, BTG-10, BTG-15 corresponding to their various graphene contents of 1 wt%, 5 wt%, 10 wt%, 15 wt%, respectively) were synthesized.

### 2.2. Characterization

The morphology, structure and element distribution of the black TiO_2_/graphene composites were examined by high-resolution transmission electron microscope (HRTEM, Tecnai G2 20, Hillsboro, OR, USA). The crystal structure of composites was detected by X-ray diffraction (XRD) on Rigaku D/MAX-2400 (Tokyo, Japan). In order to confirm the chemical compositions and band status of the composites, XPS spectrum was characterized on AXIS-Ultra DLD-600 W (Manchester, UK). The state of the graphene was characterized by Raman spectroscopy with LabRAM HR800 (Piscataway, NJ, USA) using laser excitation at 532 nm. In order to examine the light absorption range of the composites, UV–Vis absorption spectra examinations were performed on a Shimadzu UV-3600 Plus (Tokyo, Japan) UV-VIS-NIR Spectrophotometer.

The photocatalytic activity of the composites was determined in the decomposition of the methyl blue (MB). The 10 mg/L MB solution was prepared to test the visible light catalytic performance of the as-synthesized composites (BTG). A sample of 10 mg BTG was added to 100 mL MB solution and stirred in the dark for 30 min to achieve adsorption/desorption equilibrium. The visible light irradiation of the photocatalysis experiment was from a 300 W halogen tungsten lamp with a cut-off filter (λ > 420 nm). The reaction solution was controlled at 20 °C with a water-cooling system. In the photocatalysis experiment, 3 mL of reaction solution was taken every 10 min, and the catalyst was removed by centrifugation (10,000 rpm). The concentrations of residual MB were analyzed by the absorption band maximum (660 nm).

## 3. Results and Discussion

The XRD patterns of TiO_2_/graphene composites before and after the hydrogenation process are shown in Figure 2A. The XRD patterns of the TiO_2_/graphene composites before the hydrogenation process show only some diffuse peaks, indicating that the composites before hydrogenation are amorphous. After the hydrogenation process, the peaks occur at 25.28° (101), 37.80° (004), 48.05° (200), 53.89° (105), 55.07° (211), 62.69° (204), 68.93° (116), 70.31° (220), and 75.03° (215) (Figure 2A(b)), corresponding to the diffractions of anatase TiO_2_ (JCPDS 21-1272) [19,21]. The results indicate that the samples after surface hydrogenation were crystallized from amorphous to anatase structure, and the average crystal size was approximately 21 nm calculated by Scherrer formula, in agreement with TEM observation.

The structure of black TiO_2_/graphene composites can also be characterized by Raman spectra. The Raman spectra of the BTG with various graphene contents are shown in Figure 2B. The three bands at around 1365 cm^−1^ (D band), 1580 cm^−1^ (G band) and 2700 cm^−1^ (2D band) correspond to graphene [22]. For all the BTG samples, the Raman peaks occurred at around 156 cm^−1^ (Eg(1)), 406 cm^−1^ (B1g(1)), 523 cm^−1^ (A1g + B1g(2)), and 646 cm^−1^ (Eg(2)), which matched with the characteristic peaks of anatase TiO_2_ [23]. Compared with the characteristic peaks of anatase TiO_2_, the Eg(1) mode shifted from 144 cm^−1^ of bare bulk TiO_2_ to 156 cm^−1^ of BTG. The shift toward high frequency indicated the ultra-dispersed characteristics of the TiO_2_ nanoparticles and their combination with graphene, and the disappearance of the graphene 2D band in the BTG may be attributed to the composite of graphene and TiO_2_. From the Raman analysis, the characteristic peaks of TiO_2_ and graphene appeared in the spectra of BTG, indicating that the black TiO_2_/graphene composites were successfully synthesized.

The XPS spectra of the black TiO_2_/graphene composites are shown in Figure 3A. The characteristic peaks of C 1s, Ti 2p, and O 1s were present at 284.6, 457.8, and 529.7 eV, respectively. In the Ti 2p XPS spectrum (Figure 3B), the Ti 2p_3/2_ and Ti 2p_1/2_ of TiO_2_ were revealed at 457.8 and 463.3 eV. The Ti 2p_3/2_ peak of the BTG shifted from 458.6 eV to a lower binding energy corresponding to the presence of a high Ti^3+^ concentration [24,25,26,27]. All Ti 2p spectra were symmetrical on the low energy side, indicating that TiO_2_ was not doped with carbon. The curve fit of C 1s spectra of BTG is shown in Figure 3C. The peak at 284.5 eV was ascribed to the C=C/C-C bond, indicating the presence of graphene. The weak peak at 286.5 confirmed the presence of the C-O bond. In addition, there was no Ti-C peak observed in Figure 3B,C, which confirms that graphene does not exist as a dopant in BTG composites. The curve fit of O 1s spectra of BTG is shown in Figure 3D. The peak at 529.6 and 532.2 eV were ascribed to Ti-O and C-OH bonds. In Figure 3C,D, the appearance of C-O and C-OH bonds indicated the existence of a bond between carbon and oxygen atoms in BTG. This phenomenon is attributed to the oxidation of graphene since TiO_2_ is a well-known catalyst.

The selected area electron diffraction (SAED) pattern is shown in the inset of Figure 4A. The (101), (004), (200), (204) and (105) diffraction rings detected in the SAED pattern indicated that TiO_2_ in the composites was anatase structure. The SAED result was consistent with the XRD characterization, which indicated that the TiO_2_ in the composite was anatase structure with excellent photocatalytic activity. Figure 4B shows the HRTEM image of black TiO_2_/graphene composites. It can be seen that the size of individual TiO_2_ nanocrystals was approximately 15 nm in diameter. There was a disordered surface layer surrounding the TiO_2_ nanocrystal, as shown by the dotted red circle in Figure 4B. The thickness of the disordered layer is ~1 nm, which is consistent with the black TiO_2_ reported by Chen et al. [18]. The disordered layer surrounding the TiO_2_ nanocrystal was created by hydrogenation, which caused a significant color change and enhancement of visible light photocatalytic activity. The schematic diagram of the sample color change (from blue to black) after hydrogenation is shown in Figure 1. The inset in Figure 4B shows that the interplanar spacing was 3.58 Å, corresponding to the (101) plane of anatase TiO_2_. The energy dispersive X-ray (EDX) elemental mappings of Ti, C, O taken from the STEM image of Figure 4C are given in Figure 4D–F, respectively. It can be seen from the figures that the Ti and O elements were uniformly aggregated and dispersed on the C element of graphene, which was consistent with the TEM result of Figure 4A. The results further demonstrate the successful assembly of TiO_2_ on graphene in the black TiO_2_/graphene composites.

Figure 5A shows the UV–vis diffuse reflectance spectra (DRS) of the black TiO_2_/graphene composites. The presence of graphene significantly improved the visible light absorption of the black TiO_2_/graphene composites. The visible light absorption intensity of the composites was enhanced with increasing graphene content. In order to characterize the band gaps of the composites, the Kubelka–Munk function (F(R∞)·E)1/n versus the energy of light (E = hv) is shown in Figure 5B. For an indirect transition of anatase TiO_2_, *n* = 2 will give the best linear fit. As the graphene content increased from 1% to 15%, the band gaps were estimated roughly to decrease from 3.05 to 2.94 eV. It is well known that the band gap energy of anatase TiO_2_ is 3.2 eV. The band gap around 3 eV of the composites was lower than that of anatase TiO_2_, which was attributed to the self-doping of Ti^3+^. Moreover, the composites had enhanced light absorption in the range of visible light, which was consistent with the darker sample color with increasing graphene content. The results suggest that both graphene combination and self-doping of Ti^3+^ play a crucial role in the photocatalytic activity of the composites. Figure 5C illustrates the normalized MB concentration in the degradation solution as a function of visible light irradiation time. After a visible light irradiation time of 60 min, 95, 98, 99 and 96% of MB was decomposed in the presence of the BTG-1, BTG-5, BTG-10 and BTG-15, respectively. A comparative experiment without catalyst during visible light irradiation exhibited only 15% of MB decomposition. The photocatalytic process follows first-order kinetics, c = c0exp(-kt), where c_0_ and c are the MB concentration before and after visible light irradiation, respectively. The k value in the formula represents the photocatalytic reaction rate. Through fitting calculation, the photocatalytic reaction rates k for BTG-1, BTG-5, BTG-10 and BTG-15 were determined to be 2.88, 3.61, 3.99 and 3.23 h^−1^, respectively. The BTG-10 exhibited the highest photocatalytic activity. To determine the recyclability of the composites, the BTG-10 was recycled under several visible light irradiation cycles. As shown in Figure 5D, the degradation rate of MB still reached 90% after five cycles in the presence of the BTG-10, indicating that the catalyst had good stability.

Since BTG-10 had the best catalytic activity of the TiO_2_/graphene composites, BTG-10 was selected for valence band (VB) XPS analysis in Figure 6A. The VB of BTG-10 is located at 2.68 eV, which is lower than the commonly used TiO_2_ (3.0 eV). The insets in Figure 6A show the energy band diagrams. According to the energy band model, the conductance band (CB) can be calculated by CB = VB − Eg, where Eg represents the energy of the band gaps. The Eg of BTG-10 was estimated to be 3.0 eV by Figure 5B, and the CB of BTG-10 was calculated as −0.32 eV. The results suggest that the disordered surface layer surrounding the TiO_2_ nanocrystal introduced by hydrogenation can upshift both the VB and CB edge of the TiO_2_/graphene composites. According to the energy band analyses, the VB of the BTG-10 was higher than that of the O_2_/H_2_O, and the CB was lower than H^+^/H_2_, potentially suggesting that the composites have attractive potential for applications in environmental and energy issues.

The photocatalytic mechanism of the black TiO_2_/graphene composites is shown in Figure 6B. The disordered surface layer introduced by hydrogenation narrows the band gap of the composites, which improves the optical absorption properties. Consequently, the electrons in the VB can easily transit to the CB of TiO_2_ under visible light irradiation. It is well known that the graphene to which the nano-sized black TiO_2_ attached has good electrical conductivity [28]. Therefore, electrons will be transferred to graphene instead of CB, which is conducive to reducing the opportunities of electron-hole recombination and enhancing photocatalytic activity. However, the graphene itself has no photocatalytic activity, and excessive graphene will hinder the absorption of photons by TiO_2_. This hinder effect is the reason why BTG-10 had a higher catalytic activity than BTG-15.

## 4. Conclusions

A series of black TiO_2_/graphene composites with different graphene contents were successfully synthesized. In the sol–gel process, TiO_2_ was generated in situ on the surface of graphene from TBOT as a titanium source. The good conductivity of graphene is beneficial for eliminating the opportunity for photogenerated electron-hole recombination. In the hydrogenation process, self-doping Ti^3+^ was introduced, accompanying the crystallization of amorphous TiO_2_. The narrowed bandgap (2.94~3.05 eV) caused by self-doping Ti^3+^ enhanced the visible light absorption. Moreover, the nanostructured black TiO_2_-graphene composites showed enhanced visible light photocatalytic activity in methyl blue degradation. The sample with 10 wt% graphene showed the highest photocatalytic activity and good stability. The incorporation of black TiO_2_ caused by hydrogenation and graphene composite expanded the light absorption range and reduced the recombination of photogenerated electron holes, both of which enhanced the capacity of photodegrading organic dyes. Therefore, this work is expected to open up a new way for the synthesis of black TiO_2_/graphene composites, and its high photocatalytic activity proves that the black TiO_2_/graphene composites have attractive potential for applications in environmental and energy issues.

## Figures and Tables

**Figure 1 materials-15-03336-f001:**
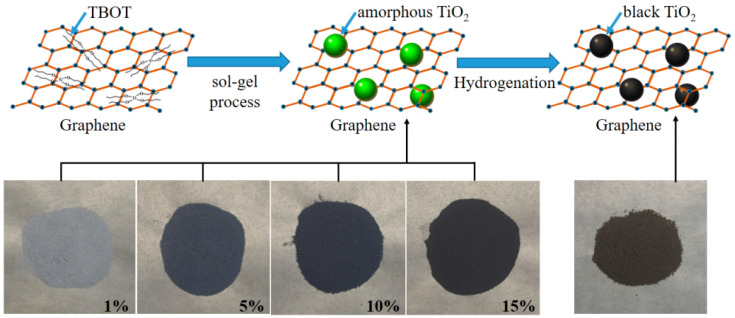
Schematic diagram showing synthetic procedure of black TiO_2_/graphene composites.

**Figure 2 materials-15-03336-f002:**
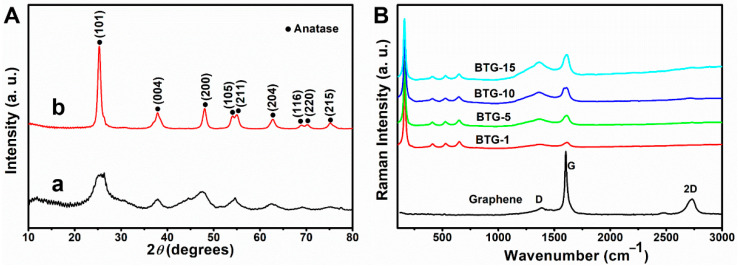
(**A**) XRD patterns of TiO_2_/graphene composites (a) before and (b) after hydrogenation process. (**B**) Raman spectra of black TiO_2_/graphene composites with different graphene contents.

**Figure 3 materials-15-03336-f003:**
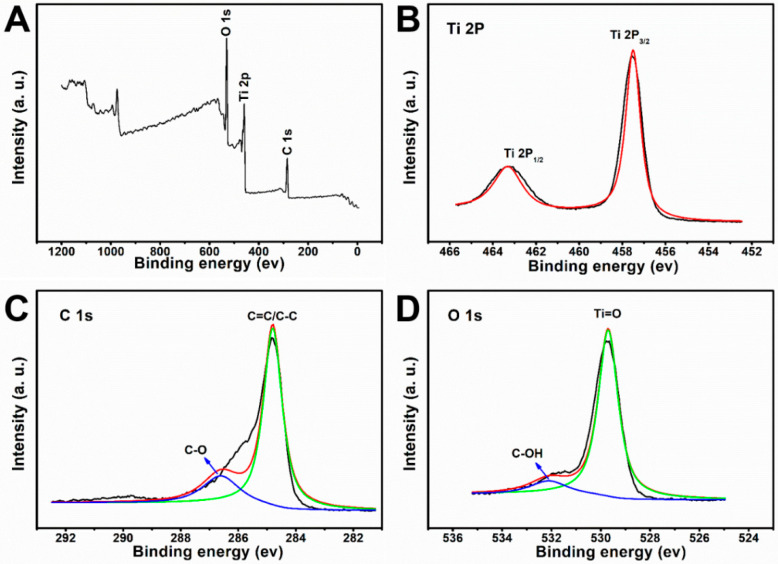
XPS spectra of black TiO_2_/graphene composites. (**A**) Full survey, (**B**) Ti 2p spectra. (**C**) C 1s spectra. (**D**) O 1s spectra.

**Figure 4 materials-15-03336-f004:**
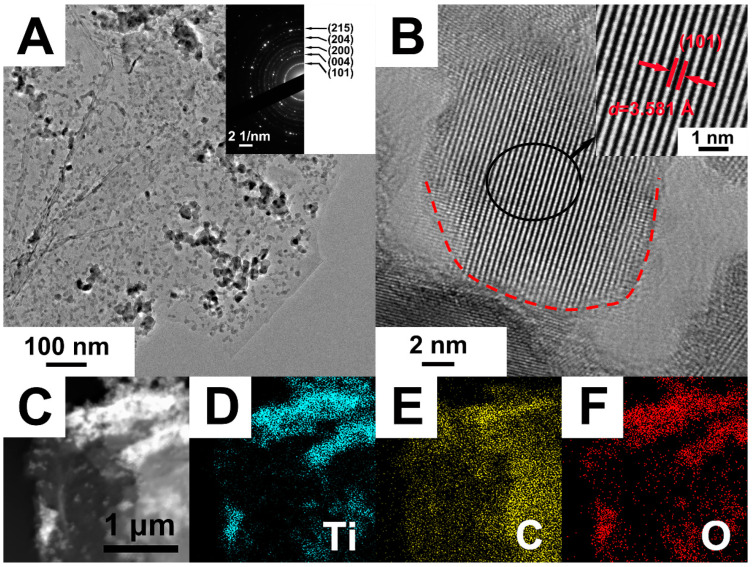
(**A**) TEM, (**B**) HRTEM and (**C**) STEM images of black TiO_2_/graphene composites. Elemental mapping of (**D**) titanium, (**E**) carbon and (**F**) oxygen taken from the whole area of (**C**).

**Figure 5 materials-15-03336-f005:**
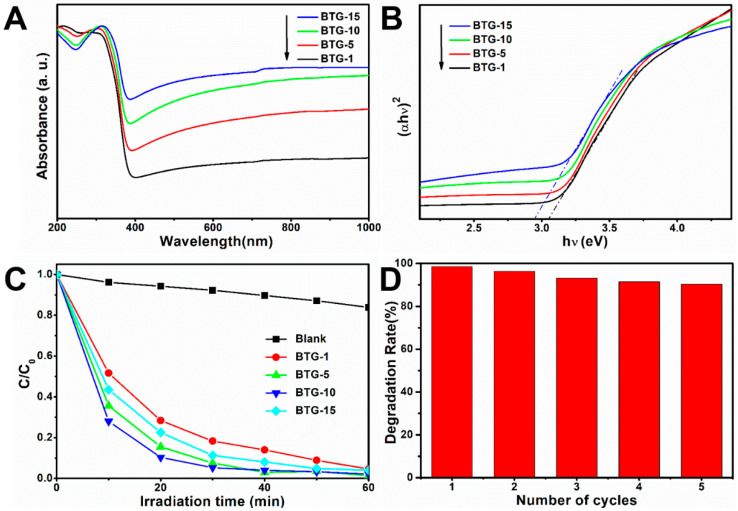
(**A**) UV–vis diffuse reflectance spectra of black TiO_2_/graphene composites, (**B**) the Kubelka–Munk function versus the energy of light, (**C**) photocatalytic degradation of methylene blue (MB) under visible light (λ > 420 nm), and (**D**) cycle test of the samples’ degradation of MB.

**Figure 6 materials-15-03336-f006:**
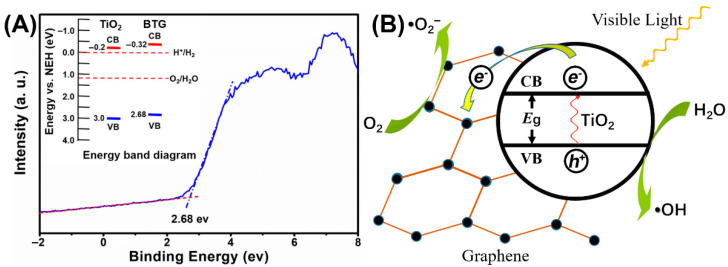
(**A**) The valence band XPS spectra of BTG-10, and (**B**) the proposed photocatalytic mechanism of BTG-10.

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
