# Peer review of "Enhanced Photocatalysis of Black TiO_2_/Graphene Composites Synthesized by a Facile Sol–Gel Method Combined with Hydrogenation Process"

_materials, 2022, doi:10.3390/ma15093336_

Round 1

Reviewer 1 Report

Dear Authors, 

Authors reported ´´Enhanced photocatalysis of black TiO2/graphene composites synthesized by a facile sol-gel method combined with hydrogenation process´´. The manuscript is interesting and can be considered for publication in Materials after solving these issues:

  1. Page: 1,  Line: 38: Also TiO2 can form composite with g-C3N4 (this sentence should be added with this citation: Sustain. Energy Fuels. 3 (2019) 2907. Line 41-44: how the formation of the composite structure can force the photocatalytic performance, authors should explain how.
  2. Page: 2: whats is the difference between your research and that done by Chen et al [17]. Further details should be added from Chen's paper, and compared to your results, what is new?. How the hydrogenation process can convert the amorphous TiO2 to the black one?
  3. page: 3: Line 94: C should be written as °C, 2-theta degrees in Line 103-104: should be written with °. Line 113: TiO2: 2: should be written in lowercase. Line: 114: cm-1: -1: should be written in uppercase. Why there are no diffraction peaks related to the graphene in the composite?
  4. Page: 4: Authors should return to the following literature for a better explanation of the XPS of Ti3+ self doped TiO2: Appl Catal B: Environ: 242 (2019) 92-99. (Line 127-129).
  5. According to Fig. 5a: BTG-15 has the lowest bandgap energy, but the BTG-10 has the highest activity? Authors should explain that in detail, and it will be better if some experiments that revealed the charge separation efficiency like photocurrent, PL, or others can be conducted. 
  6. The sentence in Line 206-208: should be rewritten again, I think here the authors missed the word lower before the redox potential of hydrogen? (Line 210-211): Authors claimed that the disordered surface layer introduced by hydrogenation narrowed the bandgap, how?
  7. It will be also good if the authors add scavenger curves to see what are the active oxygen species responsible for the degradation reaction.

Kindest regards.

Author Response

Reply to the reviewer’s comments and revision details

Manuscript title: Enhanced photocatalysis of black TiO2/graphene composites synthesized by a facile sol-gel method combined with hydrogenation process

Manuscript ID: materials-1694094

Journal: Materials (ISSN 1996-1944)

Dear Reviewers,

Thanks for your time and constructive suggestions on making this paper better. We have carefully revised our manuscript by considering the reviewer’s comments. All the revisions are highlighted in the revised manuscript using the "Track Changes". We sincerely hope that we have addressed your concerns.

Reviewer 1

Comment: Page: 1, Line: 38: Also TiO2 can form composite with g-C3N4 (this sentence should be added with this citation: Sustain. Energy Fuels. 3 (2019) 2907. Line 41-44: how the formation of the composite structure can force the photocatalytic performance, authors should explain how.

Reply: We replaced “In recent years, some carbon nanomaterials, such as carbon nanotubes and graphene with excellent electrical conductivity, are recognized as attractive composite materials for improving the photocatalytic activity of TiO2 [11, 12]” in the original version has been revised as “In recent years, some nanomaterials, such as carbon nanotubes, g-C3N4 and graphene with excellent electrical conductivity, are recognized as attractive composite materials for improving the photocatalytic activity of TiO2 [11-13]” from line 37 to line 40 on page 1 in the revised manuscript. Reference [1] and were added as Reference [13] in the revised version.

The improvement of photocatalytic performance caused by composite structure was explained as “Therefore, the contact between TiO2 and graphene can significantly pro-mote the recombination of photogenerated electron-hole, thereby improving photocatalytic performance” from line 41 to line 43 on page 1 in the revised manuscript.

Comment: Page: 2: what is the difference between your research and that done by Chen et al [17]. Further details should be added from Chen's paper, and compared to your results, what is new?. How the hydrogenation process can convert the amorphous TiO2 to the black one?

Reply: Compared with Chen's work, the amorphous TiO2 raw material prepared by sol-gel method in our work can reduce the experimental conditions of hydrogenation, and we prepared TiO2-graphene composites instead of pure TiO2.

The hydrogenation process convert the TiO2 black in color was explained in Fig. 5 from line 167 to line 170 on page 5 in the revised manuscript.

Comment: page: 3: Line 94: C should be written as °C, 2-theta degrees in Line 103-104: should be written with °. Line 113: TiO2: 2: should be written in lowercase. Line: 114: cm-1: -1: should be written in uppercase. Why there are no diffraction peaks related to the graphene in the composite?

Reply: Thanks for your careful examination on making this paper better. The mistakes have been corrected in the revised manuscript. And we checked the whole paper again.

The absence of graphene peaks in the XRD spectrum is due to the high stripping degree of the graphene (Strem Chemicals, Inc.) we used, which is generally less than 10 layers.

Comment: Page: 4: Authors should return to the following literature for a better explanation of the XPS of Ti3+ self doped TiO2: Appl Catal B: Environ: 242 (2019) 92-99. (Line 127-129).

Reply: The explanation of the XPS of Ti3+ self doped TiO2 has been revised as “In the Ti 2p XPS spectrum (Fig. 3(B)), the Ti 2p3/2 and Ti 2p1/2 of TiO2 are revealed at 457.8 and 463.3 eV. The Ti 2p3/2 peak of BTG shifts from 458.6 eV to a lower binding en-ergy corresponds to the presence of a high Ti3+ concentration [24-27].” from line 127 to line 131 on page 4 in the revised manuscript. Reference [2] and were added as Reference [27] in the revised version.

Comment: According to Fig. 5a: BTG-15 has the lowest bandgap energy, but the BTG-10 has the highest activity? Authors should explain that in detail, and it will be better if some experiments that revealed the charge separation efficiency like photocurrent, PL, or others can be conducted.

Reply: The reason why BTG-10 has higher catalytic activity than BTG-15 is explained as “However, the graphene itself has no photocatalytic activity, and excessive graphene will hinder the absorption of photons by TiO2. This hinder effect is the reason why BTG-10 has a higher catalytic activity than BTG-15.” from line 218 to line 220 on page 6 in the revised manuscript.

Comment: The sentence in Line 206-208: should be rewritten again, I think here the authors missed the word lower before the redox potential of hydrogen? (Line 210-211): Authors claimed that the disordered surface layer introduced by hydrogenation narrowed the bandgap, how?

Reply: We replaced “According to the energy band analyses, the VB of the BTG-10 is higher than that of the O2/H2O, and the CB is H+/H2 potential” in the original version has been revised as “According to the energy band analyses, the VB of the BTG-10 is higher than that of the O2/H2O, and the CB is lower than H+/H2 potential” from line 208 to line 209 on page 6 in the revised manuscript.

It is well known that the band gap energy of anatase TiO2 is 3.2 eV. The band gap around 3 eV of the BTG composites is lower than that of anatase TiO2, which is attributed to the disordered surface layer.

Comment: It will be also good if the authors add scavenger curves to see what are the active oxygen species responsible for the degradation reaction.

Reply: Thanks for your constructive suggestions, we will conduct scavenger study of the materials in the future.

References

[1] M. Ismael, Y. Wu, A mini-review on the synthesis and structural modification of gC 3 N 4-based materials, and their applications in solar energy conversion and environmental remediation, Sustainable Energy & Fuels, 3 (2019) 2907-2925.

[2] J. Pan, Z. Dong, B. Wang, Z. Jiang, C. Zhao, J. Wang, C. Song, Y. Zheng, C. Li, The enhancement of photocatalytic hydrogen production via Ti3+ self-doping black TiO2/g-C3N4 hollow core-shell nano-heterojunction, Appl. Catal., B 242 (2019) 92-99.

Reviewer 2 Report

The present work aims to study an improved photocatalysis method for synthesizing black TiO2/graphene composites using the sol-gel method combined with the hydrogenation process. Overall the work has great potential, however, some improvements are needed. Below are my enhancement suggestions.

1) Grammatical and typing errors must be corrected.

2) It is necessary to add a paragraph in the introduction showing the relevance of the present work.

3) Improve the quality of figures 2, 3, 5, and 6.

4) The Results section should be called Results and Discussion. This section needs further comparisons with works in the literature, in addition to a more detailed description of physical processes. The way it is written is just too many characterizations being described in a poor way, without a deeper scientific discussion.

5) The references are not in accordance with the journal's standard, and there are few recent references.

After these modifications, the manuscript can be published.

Author Response

Reply to the reviewer’s comments and revision details

Manuscript title: Enhanced photocatalysis of black TiO2/graphene composites synthesized by a facile sol-gel method combined with hydrogenation process

Manuscript ID: materials-1694094

Journal: Materials (ISSN 1996-1944)

Dear Reviewers,

Thanks for your time and constructive suggestions on making this paper better. We have carefully revised our manuscript by considering the reviewer’s comments. All the revisions are highlighted in the revised manuscript using the "Track Changes". We sincerely hope that we have addressed your concerns.

Reviewer 2

Comment: 1) Grammatical and typing errors must be corrected.

Reply: We checked the whole paper again.

In addition, we have made some revisions to improve English.

2) It is necessary to add a paragraph in the introduction showing the relevance of the present work.

Reply: The relevance of the present work was added as “There are still some issues for the TiO2-graphene composites, such as their low visible light utilization [15, 16]. In 2011, Chen et al. synthesized black TiO2 nanoparticles with a long wavelength absorption and substantial visible light photocatalytic activities by hydrogenation method [17]. These remarkably changed properties make the black TiO2/graphene composites a promising candidate to develop the photocatalytic perfor-mance.” from line 44 to line 49 on page 2 in the revised manuscript.

3) Improve the quality of figures 2, 3, 5, and 6.

Reply: The quality of figures 2, 3, 5, and 6 have been improved.

4) The Results section should be called Results and Discussion. This section needs further comparisons with works in the literature, in addition to a more detailed description of physical processes. The way it is written is just too many characterizations being described in a poor way, without a deeper scientific discussion.

Reply: The section 3 was named “Results and Discussion”. And we add more discussion information.

5) The references are not in accordance with the journal's standard, and there are few recent references.

Reply: The reference format has been modified to the journal format and some new references have been added.
